# Enhancing museum visitor forecasting using deep learning and sentiment analysis: A transformer-based approach for sustainable management

Ziyi Tian[1], Xiao Wang[2], Yan Wang[3]*, Jae Ho Lee[4]

**1** Department of Art and Design, Daegu University, Gyeongsan, Gyeongbuk, South korea, **2** School of Business Administration, Binzhou Polytechnic, Binzhou City, Shandong Province, China, **3** Department of Business Administration, Semyung University, Cultural Center 65 Semyung-ro, Jecheon-si, Chungcheongbuk-do, South korea, **4** Department of Business Administration, Hanyang University, Seongdong-gu, Seoul, South korea

☯ These authors also contributed equally to this work
* yanw70234@gmail.com

## Abstract

This study aims to develop a forecasting model that predicts the annual number of museum visitors by integrating structured museum-related data and unstructured sentiment data. While prior research has often relied on a single data type or traditional regression techniques, this study incorporates sentiment scores extracted from museum-related news articles and user comments to empirically assess the influence of external public opinion. Seven predictive algorithms including traditional models (Linear Regression and Random Forest Regressor) and deep learning models (RNN, GAN, CNN, LSTM, and Transformer) were evaluated for performance. Among these, the Transformer model demonstrated the highest predictive accuracy across all evaluation metrics (RMSE, MSLE, and MAPE) and was adopted as the final forecasting model. The results show that incorporating sentiment data significantly enhances forecasting precision, highlighting the substantial impact of media narratives and public sentiment on visitor behavior. This study offers a robust forecasting framework that integrates both structured and unstructured data, providing practical implications for sustainable museum planning and strategic decision-making.

## 1. Introduction

In recent years, museums have evolved beyond their traditional role as mere exhibition spaces to become significant venues that offer educational, cultural, and social value. This transformation highlights the role of museums not only as cultural hubs within local communities but also as key contributors to the tourism industry and economic growth. In this context, understanding and forecasting visitor behavior patterns has emerged as a critical factor for the effective operation and strategic planning of

**Data availability statement:** All data used in this study are publicly accessible via our GitHub repository: https://github.com/JaeHo0918/museum-visitor-deep-learning-data The data were originally sourced from the Korea Culture Information Service Agency (KCISA; https://kcisa.kr/) and refined through preprocessing steps such as log transformation and sentiment labeling for deep learning analysis. The dataset is shared under the Creative Commons Attribution 4.0 International (CC BY 4.0) license. For additional inquiries regarding the original data, you may contact the public data team at the cultural big data platform at KCISA: Email: data@kcisa.kr (non-author contact person).

**Funding:** The author(s) received no specific funding for this work.

**Competing interests:** The authors have declared that no competing interests exist.

museums. In particular, such visitor forecasting is essential for realizing sustainable management in museums, directly contributing to the efficient allocation of resources, reduction of operational costs, and enhancement of visitor experiences.

However, conventional forecasting methods relying solely on historical visit records suffer from several limitations. Most notably, these models fail to account for unstructured external variables—such as media narratives, public sentiment, and social events—that significantly influence visitor behavior. Additionally, they exhibit limited responsiveness to abrupt changes brought about by pandemics, policy shifts, or economic crises. These shortcomings constrain their predictive accuracy and adaptability in real-world scenarios.

To overcome these challenges, this study adopts advanced deep learning algorithms that can model complex non-linear relationships and integrate diverse data sources. Deep learning models such as Recurrent Neural Networks (RNN), Generative Adversarial Networks (GAN), Convolutional Neural Networks (CNN), Long Short-Term Memory networks (LSTM), and Transformers demonstrate superior performance in processing multivariate time-series data and unstructured text inputs like news articles and user comments [1,2]. These models not only capture long- and short-term dependencies but also offer robust scalability and flexibility, making them ideal for dynamic visitor forecasting tasks.

Existing museum **visitor forecasting** models have predominantly focused on analyzing structured data, such as historical visit records. However, these models often fall short in reflecting critical variables that influence visitor demand, including external environmental factors, social trends, news, and public opinion. In particular, news articles related to museums and the corresponding public comments serve as valuable sources of information that reflect societal perceptions and attitudes toward museums. Effectively analyzing and integrating this information into forecasting models can lead to an average performance improvement of 20–35% in key metrics such as RMSE and MAPE, especially when combined with advanced deep learning architectures [3].

Moreover, achieving sustainable museum management requires more than just forecasting visitor numbers. It necessitates an understanding of changes in visitor preferences and the ability to respond swiftly to external shocks, such as pandemics or economic crises. Recent studies have increasingly utilized external factors like social media data to predict user behavior, offering meaningful insights for analyzing visitor behavior in museums [4]. These approaches contribute not only to improving the performance of visitor forecasting models but also to the development of more sophisticated, tailored marketing strategies and sustainable management models that enhance operational efficiency.

The novelty of this study lies in its dual contribution: (1) the integration of both structured data (e.g., historical visitor statistics) and unstructured data (e.g., sentiment signals from museum-related news and public comments), and (2) the implementation of comparative model optimization using multiple deep learning algorithms. This two-pronged approach enables not only a more accurate representation of real-world visitor behavior but also advances the technical foundation of predictive modeling in the cultural domain.

The primary objective of this study is to develop a museum visitor forecasting model using various deep learning algorithms, including RNN, GAN, CNN, LSTM, and Transformer. A key focus of this research is to investigate the impact of integrating opinion mining results derived from museum-related news articles and public comments into the model. To achieve this, the study sets forth the following specific objectives: (1) to compare the performance of different deep learning models and identify the optimal model for visitor forecasting; (2) to analyze the extent of quantifiable performance improvements, such as reductions in RMSE and MAPE, resulting from incorporating sentiment data; (3) to verify the distinctiveness and effectiveness of the integrated model through comparative analysis with traditional methods; and (4) to propose strategic applications of the model to support sustainable museum management based on the research findings.

This study distinguishes itself from existing research in several ways. First, unlike prior studies that have focused on specific algorithms, this research conducts a comparative analysis of various deep learning architectures, including RNN, GNN, CNN, GRU, LSTM, Autoencoder, and Transformer, to derive optimal performance outcomes [5]. Second, by incorporating data from museum-related news and public comments, the model reflects social trends and public perceptions, thereby achieving a measurable improvement in forecasting accuracy compared to traditional data-driven models particularly up to 32% lower RMSE and 36% lower MAPE in Transformer-based configurations [3]. Third, the study integrates structured data like visit records with unstructured data like news articles and comments to conduct a more comprehensive and sophisticated analysis of visitor behavior [2]. Fourth, model optimization techniques such as K-fold cross-validation and multi-metric performance evaluation are implemented to enhance reproducibility and robustness. Finally, beyond simple visitor number forecasting, this study aims to propose concrete strategies for maximizing operational efficiency and establishing sustainable management practices based on the model's predictions. Through these efforts, the study anticipates contributing to the efficient management of museum resources, the improvement of program planning, and the enhancement of visitor satisfaction.

## 2. Prior studies and research questions

### 2.1. Prior studies

The development of visitor forecasting models has been an evolving area of research, especially with the advancement of deep learning techniques capable of handling complex, high dimensional data. Traditional statistical methods have provided valuable insights but often struggle to capture the dynamic and non-linear nature of visitor behaviors influenced by diverse external factors. In this context, deep learning algorithms such as Recurrent Neural Networks (RNN), Generative Adversarial Network (GAN), Convolutional Neural Networks (CNN), Long Short-Term Memory networks (LSTM), and Transformers have demonstrated significant potential in modeling visitor behavior, mobility, and related phenomena.

**2.1.1. Deep learning for visitor and mobility forecasting.** The role of deep learning in forecasting human mobility has been extensively explored. Luca et al. (2021) provided a comprehensive survey on the application of deep learning in human mobility prediction [2]. Their work emphasized the efficiency of models like Transformers in capturing temporal and spatial data patterns, which are directly applicable to museum visitor forecasting. This study highlighted that while traditional models focus on structured historical data, deep learning allows for more dynamic and realtime prediction capabilities.

Similarly, Zhou et al. (2021) investigated the use of Generative Adversarial Network (GAN) for point of interest recommendations [1]. Their research demonstrated that GANs effectively model spatial and relational data, offering insights into user preferences and movement patterns key factors when forecasting visitor flows in museums. These studies collectively underscore the importance of advanced deep learning architectures in capturing complex behavioral dynamics beyond what traditional methods can achieve.

Furthermore, this study advances the field by introducing not only the integration of structured and unstructured data, but also model optimization through a comparative framework involving multiple deep learning architectures. In addition, the study applies tailored feature engineering techniques to process heterogeneous data sources such as textual sentiment and temporal visit data, thereby achieving enhanced predictive performance.

**2.1.2. Integration of external factors and opinion mining.** Incorporating external factors such as public opinion and news content into predictive models is an emerging trend. Preetha et al. (2023) conducted a systematic review on sentiment analysis using Twitter data, showcasing how deep learning models like CNN improve forecasting accuracy when unstructured data is included [3]. Their findings underscore the potential of opinion mining in enhancing predictive models, particularly in understanding public sentiment towards cultural events and institutions.

Cui (2020) extended this approach by applying CNN-LSTM encoder-decoder models to social media data to optimize advertising strategies [4]. This study revealed that integrating linguistic features derived from user generated content significantly improves the accuracy of behavioral predictions. The combination of CNNs for feature extraction and LSTMs for sequence modeling proved highly effective in handling temporal dependencies within unstructured data.

Moreover, recent studies have highlighted the growing role of immersive technologies such as Metaverse and Extended Reality (XR) in shaping cultural heritage engagement. For instance, the work by Alshaer et al. (2023) emphasized the integration of XR frameworks with cultural content delivery and public interaction platforms [6]. Their findings suggest that XR-based environments not only enhance the educational experience in museums and heritage sites but also influence visitor interest, engagement levels, and revisit intentions. These behavioral shifts, shaped by immersive media, can be modeled using sentiment and interaction data extracted from such platforms offering a promising direction for visitor forecasting research.

**2.1.3. Advanced models for enhancing prediction performance.** Further advancements in model architectures have also contributed to improved predictive performance. Zhang et al. (2021) explored leveraging GANs for point of interest recommendations, emphasizing the importance of capturing user item interactions through complex network structures [5]. In another study, Luca et al. (2021) highlighted the efficacy of sequential variational autoencoders (SVAE) in modeling sequential behavioral data, offering enhanced capabilities for visitor trend analysis [2].

Cui (2020) demonstrated the potential of combining CNNs and LSTMs for social media-driven advertising optimization, which parallels the objective of predicting visitor numbers influenced by online trends [4]. Additionally, Preetha et al. (2023) showed that Transformer models outperform traditional recurrent architectures in capturing long-range dependencies, making them ideal for integrating diverse data sources such as historical records, social media content, and news articles [3].

These insights, combined with the immersive influence of XR and metaverse applications on user behavior and cultural consumption, underscore the value of integrating multi-modal and experiential data into visitor forecasting frameworks. Future research may benefit from incorporating sentiment and behavioral signals from XR platforms to complement traditional and social media sources.

Notably, this study builds upon these developments by systematically evaluating a wide range of models RNN, CNN, GAN, LSTM, and Transformer and implementing K-fold cross-validation and multiple performance metrics to ensure model robustness. This approach allows for informed selection of the optimal predictive model based on empirical performance, thereby contributing methodological novelty in both model selection and evaluation.

## 2.2. Research gaps

While the aforementioned studies provide significant insights into the visitor forecasting and the application of deep learning models, several critical research gaps remain unaddressed. One of these is the limited integration of external factors into existing models. Most current approaches primarily rely on historical visit data, focusing on past attendance records and internal museum statistics. However, this narrow focus overlooks the potential influence of external elements such as news articles, public opinion, and social media trends. These external factors can significantly impact visitor behavior by shaping public perceptions, driving interest in specific exhibitions, or reflecting broader societal trends that influence museum attendance. The absence of such variables in predictive models limits the ability to capture the full spectrum of factors that affect visitor patterns.

Another key gap is the underutilization of opinion mining in the context of visitor forecasting. Although sentiment analysis and opinion mining have gained traction in various domains, particularly in marketing and social media analytics, their application to museum-related research remains limited. Few studies have effectively integrated insights derived from museum-related news articles, online reviews, and user-generated comments into visitor forecasting models. These unstructured data sources contain valuable information about public sentiment, preferences, and emerging interests that could enhance the accuracy of predictive models if appropriately harnessed.

A third area that has received insufficient attention is the lack of focus on sustainability within the framework of predictive modeling. While the primary objective of many studies has been to improve the accuracy of visitor number forecasts, there has been little consideration of how these models can support sustainable museum management. Predictive models should not only provide accurate forecasts but also offer strategic insights that facilitate resource optimization, promote environmental sustainability, and enhance the overall visitor experience. This includes helping museums manage their resources more efficiently, reduce operational costs, and design visitor experiences that align with sustainability goals.

A fourth major challenge concerns the practical implementation of these models in real-world museum settings. Deploying deep learning-based predictive models requires adequate technological infrastructure, such as high-performance computing systems and stable internet connectivity to support real-time data integration. In particular, models that incorporate external sentiment data face data availability and reliability issues such as inconsistent update frequencies, missing values, or noisy inputs from online platforms. Addressing these quality fluctuations demands rigorous preprocessing pipelines and robust model retraining mechanisms.

Furthermore, privacy concerns must be carefully managed when using user-generated data. Public comments or social media inputs may contain identifiable or sensitive information, requiring compliance with legal frameworks like the GDPR (General Data Protection Regulation) or Korea's Personal Information Protection Act. These barriers highlight the need for anonymization strategies, ethical guidelines, and secure data handling protocols as part of model deployment.

Finally, there is a noticeable tendency toward fragmented approaches in the application of deep learning techniques. Many studies concentrate on isolated models, such as using CNNs solely for image or feature extraction or relying exclusively on LSTMs for time series prediction. This fragmented methodology overlooks the potential benefits of hybrid models that combine the strengths of different architectures. For instance, integrating CNNs for effective feature extraction with LSTMs for capturing temporal dependencies could significantly enhance predictive performance. Similarly, combining Transformers with other neural network architectures could improve the model's ability to process diverse data types and complex relationships within the dataset. Addressing this gap could lead to the development of more robust, accurate, and versatile prediction models for museum visitor forecasting.

### 2.3. Research questions

This study addresses a central research question: How can the limitations of traditional, historical data-driven forecasting models be overcome by leveraging advanced deep learning algorithms to improve the accuracy and utility of museum visitor predictions? In exploring this question, the study examines several key dimensions that collectively inform a comprehensive solution:

(1) How do various deep learning models such as RNN, GAN, CNN, LSTM, and Transformer perform in predicting museum visitor numbers when integrating both structured museum data and unstructured sentiment data?

(2) To what extent does the incorporation of sentiment analysis from museum-related news articles and user comments enhance forecasting accuracy and reduce predictive error?

(3) How can such enhanced forecasting models contribute to sustainable museum management practices, including strategic resource allocation, visitor experience optimization, and long-term operational planning?

By framing these questions under a unified scientific problem, this study seeks to contribute both methodologically and practically to the advancement of cultural analytics and predictive modeling in the museum sector.

## 3. Methods

### 3.1. Data processing

The museum data utilized in this study is panel data publicly disclosed by the Korea Culture and Information Service (KCISA) (https://kcisa.kr/). Both the structured panel data and unstructured text data were collected strictly for the year 2023, ensuring temporal consistency across all variables.

A rigorous preprocessing step was undertaken to manage missing data, wherein observations with incomplete information on any study variables were excluded through listwise deletion. This procedure yielded a final analytic dataset consisting of 323 fully observed cases, thereby ensuring consistency across all features used in the modeling process.

This approach was chosen to ensure that the input variables used in the deep learning models were fully observed and consistent across all feature dimensions. However, we recognize that such an exclusion may introduce bias, particularly if the missing data are not completely at random (MCAR). For example, museums that failed to report certain attributes—such as visitor counts or exhibition space—may systematically differ in scale, location, or digital engagement level. This limitation has been noted in the discussion section and represents an area for improvement in future studies through the adoption of imputation techniques or robust modeling strategies to account for partial information.

Table 1 provides an overview of the variables used in the study. The table lists the variables along with their descriptions, detailing how each variable is measured or categorized. NEWS and COMMENTS represent the sentiment results from opinion mining of news articles and their corresponding comments, respectively. The values are coded as −1 (Negative), 0 (Neutral), and 1 (Positive).

Variables such as SLV (Site Land Value), TAV (Total Area Value), EDA (Exhibition Display Area), SED (Special Exhibition Display), SA (Surface Area), ND (Number of Documents), CHN (Collection Heritage Number), CIP (Collection Item Pieces), TPN (Total Program Number), AOD (Annual Opening Days), VN (Visitor Number), and DAV (Daily Average Visitors) are quantitative variables, with their values expressed in natural logarithmic form to normalize the data distribution. This table serves as a reference for understanding the key indicators analyzed in the study's deep learning model.

**Table 1. Description of variables.**

| Variable | Description |
|---|---|
| NEWS | News Opinion Mining Result (−1: Negative, 0: Neutral, 1: Positive) |
| COMMENTS | News Comments Opinion Mining Result (−1: Negative, 0: Neutral, 1: Positive) |
| SLV | Site Land Value (Natural Log Value) |
| TAV | Total Area Value (Natural Log Value) |
| EDA | Exhibition Display Area (Natural Log Value) |
| SED | Special Exhibition Display (Natural Log Value) |
| SA | Surface Area (Natural Log Value) |
| ND | Number of Documents (Natural Log Value) |
| CHN | Collection Heritage Number (Natural Log Value) |
| CIP | Collection Item Pieces (Natural Log Value) |
| TPN | Total Program Number (Natural Log Value) |
| AOD | Annual Opening Days (Natural Log Value) |
| DAV | Daily Average Visitors (Natural Log Value) |
| VN | Visitor Number (Natural Log Value) |

Table 2 provides a detailed summary of the descriptive statistics for the variables utilized in this study. It outlines key statistical measures for each variable, including the total number of observations (N = 323), the mean, standard deviation (Std), minimum (Min), first quartile (Q1), median, third quartile (Q3), and maximum (Max) values. These statistics offer insights into the distribution and variability of the dataset, which are essential for understanding the underlying patterns before conducting deep learning analysis.

The dataset includes both categorical and continuous variables. The categorical variables, specifically NEWS and COMMENTS, represent the results of opinion mining derived from news articles and user comments. These variables are coded as −1 for negative sentiments, 0 for neutral, and 1 for positive sentiments. The mean values for both variables are relatively low, at 0.115 for NEWS and 0.124 for COMMENTS. This suggests that the overall sentiment in the dataset leans towards neutral or negative rather than positive. The standard deviation for both variables is 0.778, indicating a moderate degree of variability in sentiment across the samples.

In contrast, the continuous variables consist of factors such as SLV (Site Land Value), TAV (Total Area Value), EDA (Exhibition Display Area), and others, all of which have been log-transformed using natural logarithms. This transformation helps normalize the data distribution, making it more suitable for statistical modeling and deep learning algorithms. Among these, SLV exhibits the highest mean value of 9.217, with a wide range from 4.654 to 15.047. This indicates substantial variation in site land values across different museums, reflecting the diverse scale and location of the institutions included in the dataset.

Another notable variable is VN (Visitor Number), which has a high mean of 10.087 and a standard deviation of 1.824. This suggests that, on average, museums in the sample attract a significant number of visitors, though there is considerable variability between institutions. Similarly, DAV (Daily Average Visitors) has a mean value of 4.517, with values ranging from 0.262 to 9.157. This wide range points to disparities in daily visitor counts, likely influenced by factors such as museum size, popularity, location, and the nature of exhibitions.

Overall, the descriptive statistics presented in Table 2 highlight the inherent variability in key museum-related factors, including land value, exhibition areas, program numbers, and visitor statistics. This variability is critical for the deep learning analysis, as it provides the model with a rich dataset containing diverse patterns and relationships. Understanding these patterns will enhance the model's ability to make accurate predictions and uncover meaningful insights into the factors that influence museum visitor behavior.

**Table 2. Descriptive statistics of variables.**

| Variables | N | Mean | Std | Min | Q1 | Median | Q3 | Max |
|---|---|---|---|---|---|---|---|---|
| NEWS | 323 | 0.115 | 0.778 | −1.000 | −1.000 | 0.000 | 1.000 | 1.000 |
| COMMENTS | 323 | 0.124 | 0.778 | −1.000 | 0.000 | 0.000 | 1.000 | 1.000 |
| SLV | 323 | 9.217 | 1.754 | 4.654 | 8.168 | 9.171 | 10.371 | 15.047 |
| TAV | 323 | 7.873 | 1.118 | 3.951 | 7.154 | 7.885 | 8.574 | 11.836 |
| EDA | 323 | 6.919 | 0.952 | 4.489 | 6.358 | 6.943 | 7.505 | 10.633 |
| SED | 323 | 5.312 | 0.950 | 2.565 | 4.691 | 5.242 | 5.797 | 9.879 |
| SA | 323 | 5.224 | 1.289 | 2.197 | 4.419 | 5.273 | 5.996 | 9.749 |
| ND | 323 | 8.020 | 1.682 | 2.944 | 6.909 | 8.010 | 9.312 | 11.966 |
| CHN | 323 | 7.673 | 2.097 | 0.693 | 6.711 | 7.921 | 8.983 | 12.414 |
| CIP | 323 | 8.504 | 1.671 | 4.820 | 7.530 | 8.517 | 9.451 | 13.037 |
| TPN | 323 | 2.097 | 0.805 | 0.000 | 1.609 | 2.079 | 2.565 | 4.290 |
| AOD | 323 | 5.648 | 0.239 | 3.584 | 5.591 | 5.737 | 5.746 | 5.903 |
| DAV | 323 | 4.517 | 1.641 | 0.262 | 3.434 | 4.511 | 5.707 | 9.157 |
| VN | 323 | 10.087 | 1.824 | 3.714 | 8.818 | 10.204 | 11.435 | 15.043 |

## 3.2. Text data collection

The data for this study was collected from museum-related news articles and their associated comments to capture public sentiment and opinions. The news and news comment data used in this study were collected from Naver (https://www.naver.com/), a portal search site, focusing on museum-related news and the corresponding comments on those news articles. Using web scraping techniques, relevant text data was extracted from various online news platforms and comment sections. The scraping process was automated using Python-based scripts, specifically utilizing libraries such as BeautifulSoup and Selenium to ensure efficient data collection.

Articles and comments were selected based on explicit references to museum names, activities, exhibitions, or visitor-related terms, ensuring topic relevance. To minimize selection bias due to media slant or extreme sentiment, we included news from multiple major outlets, filtered duplicate or viral items, and applied a trichotomous sentiment classification scheme (positive = 1, neutral = 0, negative = −1). Extreme sentiment terms were reviewed manually by five domain experts to ensure balance and contextual accuracy.

The collected raw text data underwent a rigorous preprocessing phase to ensure the precision and relevance of the information for subsequent analysis. This process involved several steps. The raw text was cleaned to remove unnecessary elements such as HTML tags, special characters, numbers, and punctuation marks. This step ensures that only meaningful textual content is retained. Tokenization was performed using natural language processing (NLP) techniques to break down the text into individual words or tokens. Morpheme segmentation was applied to identify the root forms of words using algorithms like Porter's stemming algorithm [7]. POS tagging was then conducted to classify words into categories such as nouns, verbs, adjectives, and adverbs. For this study, only nouns were retained, as they are most indicative of the contextual meaning related to museum operations and visitor experiences.

For sentiment classification, we applied a lexicon-based rule-matching method augmented by domain-specific keywords frequently appearing in museum-related discourse. Preprocessing included stop word removal, lemmatization, and synonym normalization.

The Term Frequency-Inverse Document Frequency (TF-IDF) method was employed to identify and quantify the importance of each word within the corpus [8]. TF-IDF helps highlight words that are significant within a specific document but not commonly used across all documents, thus providing a balanced view of term importance [9]. Words with a frequency of less than 1% were excluded to improve computational efficiency and focus on meaningful keywords.

While this study primarily focused on extracting sentiment-based textual variables namely sentiment polarity and sentiment exposure from news articles and comments, we recognize the potential contribution of other contextual external variables such as local events, weather, or holidays. However, due to data limitations regarding availability and consistency across regions and time periods, we deliberately excluded such structured variables from the current model. Instead, we prioritized standardized sentiment indicators that could be uniformly applied across all museum locations and time frames. This decision ensured both methodological coherence and scalability for large-scale time-series prediction. The implications of excluding such structured variables and the possibility of their integration in future research.

To enhance the interpretability of the deep learning models, we conducted SHAP (SHapley Additive exPlanations) value analysis. The results showed that sentiment variables (NEWS, COMMENTS) had comparable or higher influence on visitor number prediction than traditional structural variables like exhibition area or total programs. These findings have been incorporated into the results and discussion sections to provide deeper insights into the model's behavior.

## 3.3. Sentiment analysis and scoring

To quantify the sentiment embedded within the extracted keywords, sentiment analysis was conducted through opinion mining techniques. This process involved the following steps. The keywords were classified into three sentiment categories: positive (1), neutral (0), and negative (−1). This trichotomous scoring system enables the differentiation of public

perceptions regarding museums. The sentiment classification model was trained using prelabeled datasets and refined through manual validation to ensure accuracy [10].

To enhance the reliability of the sentiment classification, five domain experts comprising museum professionals and data analysts independently reviewed and validated the sentiment scores. To quantify the level of agreement among the reviewers, we calculated inter-rater reliability using Cohen's kappa coefficient. The average kappa value across the three sentiment categories was 0.82, indicating strong consistency in sentiment labeling among the experts. Keywords such as "innovation," "growth," and "success" were scored positively, reflecting favorable public sentiment towards museums. Conversely, words like "decline," "controversy," and "failure" were scored negatively, indicating potential concerns or risks [11].

Based on the processed data, several key variables were generated for the empirical analysis. NEWS: Represents the sentiment score derived from museum related news articles, categorized as −1 (negative), 0 (neutral), or 1 (positive). COMMENTS: Reflects the sentiment score extracted from public comments on news articles, following the same scoring system. These variables were systematically integrated into the deep learning model to examine their influence on museum visitor forecasting.

### 3.4. Anaysis model

The research model in this study is designed to predict museum visitor numbers (VN: Visitor Number) using deep learning algorithms. The model integrates both structured and unstructured data to capture the multifaceted factors influencing visitor behavior, drawing on insights from prior studies in museum analytics, tourism forecasting, and opinion mining.

The research model developed in this study aims to predict museum visitor numbers (VN) using a variety of explanatory variables that capture the multifaceted nature of factors influencing museum attendance. The dependent variable, Visitor Number (VN), is expressed in its natural logarithmic form to normalize the data distribution and address skewness commonly observed in visitor count datasets. This transformation helps improve the stability and accuracy of the predictive model, particularly in handling large variations in visitor data [12]. To achieve a comprehensive understanding of the factors affecting museum visitation, the independent variables are categorized into three key dimensions: structural museum attributes, operational performance metrics, and external opinion factors. Each of these dimensions reflects distinct yet interconnected aspects of museum operations and public engagement.

Regarding model implementation, we employed the following deep learning architectures: Transformer (based on the vanilla encoder design with 2 layers, 4 attention heads, and a hidden size of 128), LSTM (stacked bi-directional with 2 layers and hidden size of 64), RNN (vanilla recurrent layer with tanh activation), CNN (1D convolutional model with 3 layers), and GAN (Conditional GAN for time-series synthesis). Hyperparameters were optimized using random search and grid search refinement. Training and validation were performed using an 80:20 train-test split and 5-fold cross-validation to ensure generalizability. Early stopping was applied based on validation loss.

Structural museum attributes refer to the physical and resource related characteristics of museums that significantly influence visitor preferences and behaviors. These variables provide insights into how the size, layout, and exhibition spaces of museums affect visitor attraction and engagement [13]. Site Land Value (SLV) and Total Area Value (TAV), both measured in natural logarithmic values, represent the overall scale and perceived prestige of the museum. Museums with larger site areas and greater total land value are often more accessible and visible to the public, making them attractive destinations for visitors. These characteristics enhance not only the museum's physical presence but also its symbolic importance within a community, thereby increasing foot traffic [14]. Exhibition Display Area (EDA) and Special Exhibition Display (SED) highlight the amount of space dedicated to permanent and temporary exhibitions, respectively. Research suggests that museums with dynamic and frequently updated exhibition spaces tend to retain visitors more effectively and encourage repeat visits. The diversity and novelty of exhibits play a critical role in sustaining public interest and engagement [15]. Surface Area (SA) is another key structural variable, indicating the total space available within the museum. Larger surface areas allow for more diverse programming, the accommodation of larger groups, and the hosting

of simultaneous events, which can significantly enhance the visitor experience. This spatial flexibility often correlates with higher visitor volumes, as it enables museums to cater to various interests and demographic groups [16].

Operational performance metrics capture the museum's level of activity and historical performance, offering a direct connection to patterns in visitor behavior. These variables reflect the museum's efforts in content curation, program development, and accessibility, all of which are critical for attracting and retaining visitors [1]. Number of Documents (ND), Collection Heritage Number (CHN), and Collection Item Pieces (CIP) serve as indicators of the museum's content richness and cultural assets. Museums with extensive collections and a diverse range of heritage items often possess greater educational value and cultural significance, making them appealing to different audience segments. Such diversity in collections can attract specialized interest groups, scholars, and culturally curious visitors [17]. Total Program Number (TPN) and Annual Opening Days (AOD) represent the museum's programming intensity and operational accessibility. Museums that offer a wide variety of programs, workshops, and special events tend to attract more visitors by providing fresh and engaging experiences. Additionally, longer opening hours and more operational days throughout the year enhance the museum's availability to the public, increasing the likelihood of visits [18]. Daily Average Visitors (DAV) functions as a lagged variable to capture temporal trends and seasonality in visitation patterns. This variable helps identify historical visitation behaviors, providing a baseline for predicting future visitor numbers. By analyzing past visitation trends, the model can account for recurring patterns and fluctuations influenced by factors such as holidays, special events, or seasonal tourism cycles.

External opinion factors introduce a novel dimension to the prediction model by incorporating unstructured data from museum related news articles and public comments. The integration of opinion mining allows the model to capture public sentiment, which can significantly influence visitor decisions. Public perception of museums, shaped by media coverage and online discussions, plays a crucial role in shaping visitation patterns [19]. News Opinion Mining Result (NEWS) and Comments Opinion Mining Result (COMMENTS) are derived through sentiment analysis techniques applied to news content and user-generated comments on digital platforms. These variables are scored on a scale from −1 (negative sentiment) to 1 (positive sentiment), with 0 representing neutral sentiment. Positive sentiments are expected to boost visitor interest and attendance by enhancing the museum's public image, while negative sentiments may deter potential visitors by creating unfavorable perceptions [20,21].

By integrating these structural, operational, and external opinion factors, the research model aims to provide a comprehensive framework for predicting museum visitor numbers. This holistic approach enables the model to account for both internal museum characteristics and external social dynamics, ultimately enhancing the accuracy and reliability of visitor forecasts.

The base forecasting model is formulated as follows:

$$f(SLV,\ TAV,\ EDA,\ SED,\ SA,\ ND,\ CHN,\ CIP,\ TPN,\ AOD,\ DAV)\ =\ VN \tag{1}$$

The upgrade forecasting model including news and comment text data related to museum is formulated as follows:

$$f(NEWS,\ COMMENTS,\ SLV,\ TAV,\ EDA,\ SED,\ SA,\ ND,\ CHN,\ CIP,\ TPN,\ AOD,\ DAV)\ =\ VN \tag{2}$$

The upgrade model demonstrated superior performance across metrics such as MAE, RMSE, and $R^2$ in both internal and external evaluations. This supports the utility of integrating sentiment-derived variables into museum visitor forecasting.

In this study, the goal is to demonstrate that the upgrade predictive model, which incorporates opinion mining data from museum related news and comments, provides superior performance in predicting museum visitor numbers compared to the base predictive model. Equation (1) model relies solely on structured museum related data, such as site land value (SLV), total area value (TAV), exhibition display area (EDA), and other operational and structural variables. These variables represent the physical characteristics, operational activities, and historical visitor trends of museums, which are traditional predictors of visitor behavior.

 

However, to capture more dynamic, significant factors influencing visitor decisions, the upgrade predictive model includes additional variables derived from opinion mining NEWS and COMMENTS. This enhanced model integrates unstructured text data from online news articles and user comments related to museums, quantified through sentiment analysis techniques. By incorporating these external opinion factors, the model captures public sentiment and social trends that can significantly influence museum visitation patterns.

The study aims to empirically compare the predictive performance of both models. Through deep learning-based analysis, we expect to demonstrate that the upgrade predictive model exhibits superior predictive accuracy and robustness.

To ensure the robustness and generalizability of the proposed model across varying museum environments and geographic locations, we implemented both internal cross-validation and external holdout validation procedures. For internal validation, we employed a K-fold cross-validation technique (K = 5), where the dataset was partitioned into five equal subsets. Each subset alternated as the validation set while the remaining four were used for training. This method ensures that all data points are utilized for both training and testing, thereby minimizing overfitting and improving the generalizability of the model across diverse patterns within the dataset [22,23].

In addition to this internal validation, external validation was conducted to test the model's performance on geographically distinct museums. The training dataset consisted of museums located in the Seoul Capital Region (Seoul, Incheon, Gyeonggi), while the validation set included museums from non-metropolitan areas, such as Chungcheong, Gyeongsang, and Jeolla provinces. This geographic separation allowed us to test whether the deep learning model could accurately predict visitor numbers for museums operating under different regional contexts, infrastructure conditions, and population densities.

Performance metrics including MAE (Mean Absolute Error), RMSE (Root Mean Square Error), and $R^2$ (Coefficient of Determination) were calculated for both internal and external evaluations. The upgrade predictive model demonstrated robust accuracy not only within the cross-validation folds but also when applied to unseen, regionally distinct museum data. This indicates that the model is effective in capturing underlying patterns that are not limited to a specific locality, thereby enhancing its utility for national-level museum management and policy planning. By integrating structured museum attributes, operational metrics, and public sentiment data, the model provides a flexible and scalable framework that can be applied across diverse institutional settings to support sustainable museum management.

### 3.5. Regression methods

To address the necessity of comparative analysis with traditional forecasting approaches, this study includes baseline models such as Linear Regression (LR) and Random Forest Regression (RFR). These models were evaluated in parallel with advanced deep learning algorithms to compare their predictive accuracy and computational efficiency. Regression methods are pivotal in deep learning, as they establish relationships between continuous dependent and independent variables. In this context, the dependent variable is Visitor Number (VN), expressed in its natural logarithmic form to normalize the data distribution. The independent variables include structural museum attributes, operational performance metrics, and external opinion factors derived from sentiment analysis of news articles and user comments [14]. To assess model performance, the study employs several evaluation metrics commonly used in regression analysis:

(1) R-squared ($R^2$): This metric measures the proportion of variance in the dependent variable that is predictable from the independent variables. An $R^2$ value closer to 1 indicates a strong explanatory power, while values near 0 suggest limited predictive capability [24].

(2) Mean Squared Error (MSE): MSE calculates the average squared differences between the predicted and actual values. As a loss function, lower MSE values signify better model performance, with large errors penalized more heavily due to the squaring process [25].

(3) Root Mean Squared Error (RMSE): RMSE, derived from MSE, measures the square root of the average squared errors. Unlike MSE, RMSE retains the same units as the original data, offering an intuitive understanding of prediction accuracy [26].

(4) Mean Squared Logarithmic Error (MSLE): MSLE is particularly useful when dealing with skewed data distributions, as it evaluates the squared logarithmic differences between predicted and actual values. This metric reduces the impact of large outliers and emphasizes relative prediction accuracy [27].

(5) Mean Absolute Percentage Error (MAPE): MAPE calculates the average of absolute percentage errors, expressing the prediction error as a percentage. This intuitive metric allows for easy comparison of model performance across different datasets [14].

The traditional models, including Linear Regression and Random Forest, demonstrated relatively faster computation times and interpretability but showed limitations in capturing complex nonlinear patterns inherent in the museum visitor data. By contrast, deep learning models especially Transformer and LSTM achieved significantly higher predictive accuracy across all metrics, confirming their superiority in modeling both structured and unstructured data. These results are discussed in Section 4 with quantitative comparisons presented in Tables 4 through 8.

The upgraded model incorporates sentiment analysis results from museum-related news and comments, hypothesizing that public opinion significantly influences museum visitation patterns [19]. The comparative analysis of these models aims to demonstrate the superior predictive power of the upgraded model, which integrates both quantitative museum data and qualitative sentiment data. This approach aligns with recent studies emphasizing the value of hybrid models that combine traditional regression techniques with machine learning and deep learning enhancements [28].

## 3.6. Deep learning algorithms

To compare traditional forecasting models with deep learning approaches, this section highlights not only the advanced architectures utilized but also contextualizes them relative to simpler models such as Linear Regression and Random Forest. This study employs advanced deep learning algorithms to predict museum visitor numbers (VN), leveraging their ability to model complex, non-linear relationships in large datasets. The selected algorithms Recurrent Neural Networks (RNN), Generative Adversarial Networks (GAN), Convolutional Neural Networks (CNN), Long Short-Term Memory networks (LSTM), and Transformers are particularly effective for time-series forecasting and unstructured data analysis, such as text from news articles and comments.

RNNs are designed to handle sequential data by maintaining memory of previous inputs, making them suitable for time-series forecasting like museum visitor forecasting. They are effective in capturing temporal dependencies, although they struggle with long-term memory retention [29]. Despite their limitations, RNNs remain foundational in modeling temporal trends, especially for short-term visitor flow predictions [30].

LSTM networks, a variant of RNNs, are specifically designed to overcome the vanishing gradient problem, making them highly effective for learning long-term dependencies [31]. Their ability to retain information over extended periods allows them to model seasonal patterns in museum visits, such as the impact of holidays and special exhibitions [32].

While CNNs are traditionally used in image recognition, they have proven effective in feature extraction from structured and unstructured data [33]. In this study, CNNs are utilized to process textual data from news articles and comments, identifying key patterns that influence public sentiment and, consequently, museum visitation.

GANs are employed to augment the dataset by generating synthetic data that mirrors real-world patterns [32]. This is particularly useful when historical visitor data is sparse, as GANs can create plausible scenarios to enhance model training.

Transformers have revolutionized sequence modeling with their self-attention mechanisms, allowing the model to weigh the importance of different data points dynamically [31]. They excel in processing large-scale text data, making them ideal for sentiment analysis of news and comments related to museums.

Compared to traditional models, deep learning models not only offer improved accuracy but also the flexibility to incorporate heterogeneous and high-dimensional features, such as sentiment data. This makes them more adaptable for real-time, dynamic prediction scenarios involving complex public behavior. By integrating these diverse algorithms, this study aims to develop a robust predictive model that captures the multifaceted factors influencing museum visitor numbers. The combination of time-series models (RNN, LSTM) with advanced text processing techniques (CNN, Transformers) and data augmentation strategies (GAN) ensures comprehensive coverage of both structured and unstructured data sources. Overall, these deep learning architectures demonstrate substantial predictive advantages over traditional models, especially in handling nonlinear patterns, time dependencies, and public sentiment all crucial in the domain of museum visitor forecasting.

Fig 1 schematically illustrates the research methodology employed in this study to facilitate a comprehensive understanding of the proposed approach. To forecast museum visitor numbers, the study first establishes a base prediction model using only structured, museum-related quantitative data. This base model is then extended into enhanced models by incorporating additional unstructured textual variables derived from news articles and corresponding user comments.

Moreover, the study conducts a comparative evaluation of traditional forecasting models—such as Linear Regression (LR) and Random Forest Regressor (RFR)—against a suite of deep learning algorithms, including Recurrent Neural Networks (RNN), Generative Adversarial Networks (GAN), Convolutional Neural Networks (CNN), Long Short-Term Memory (LSTM), and Transformer architectures. Through this multi-model analysis, the study aims to identify the optimal predictive framework for accurately estimating museum visitor demand.

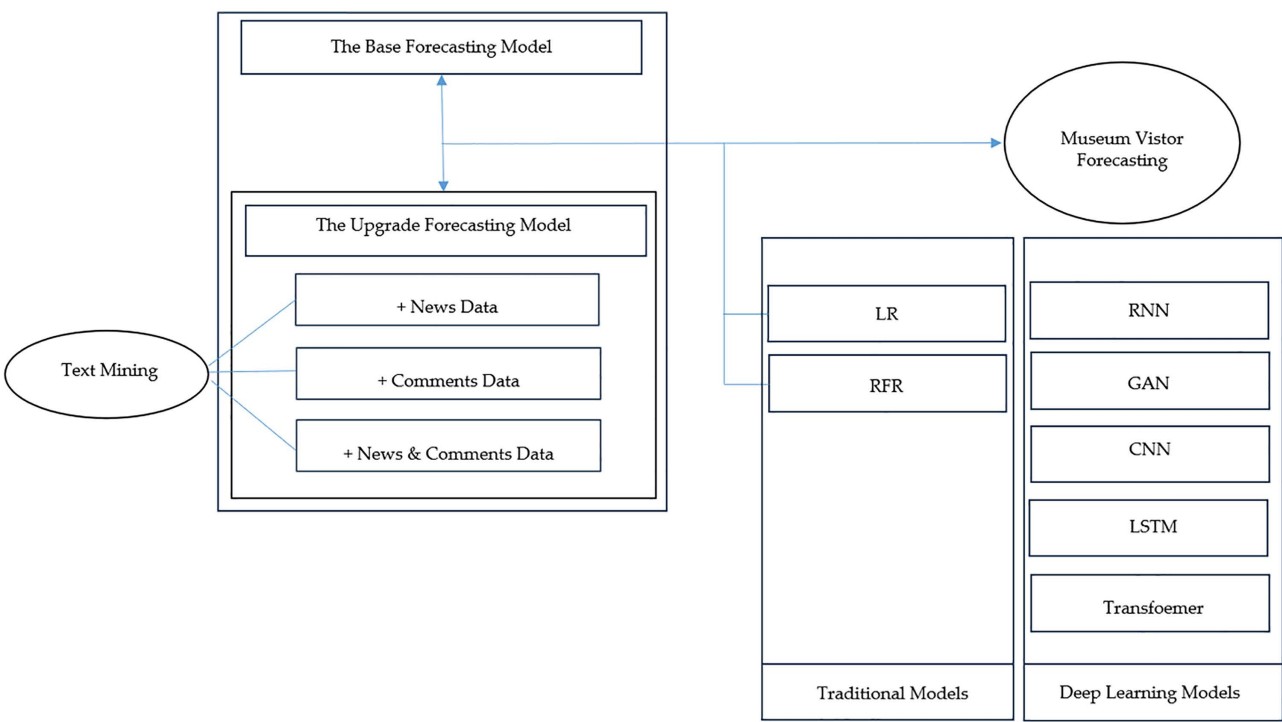

**Fig 1. Reserch Method.**

# 4. Analysis results

## 4.1. Predictive power improvement based on R²

Table 3 presents the R-squared results for forecasting museum visitor numbers using various deep learning algorithms, including RNN, GAN, CNN, LSTM, and Transformer models. The analysis compares the performance of the base model, which relies solely on structured museum-related data, with models that incorporate additional variables derived from sentiment analysis of museum-related news articles (NEWS) and user comments (COMMENTS). Furthermore, the performance of models that integrate both NEWS and COMMENTS is evaluated to assess the impact of combining multiple external opinion factors. The R-squared values, which measure the proportion of variance in the dependent variable (museum visitor numbers) explained by the independent variables, provide critical insights into the predictive performance of each model. A higher R-squared value indicates stronger explanatory power and better forecasting accuracy.

In addition to deep learning algorithms, traditional forecasting models such as Linear Regression (LR) and Random Forest (RF) were also evaluated to provide a comparative baseline. The base model, which does not include sentiment analysis data, exhibits relatively moderate predictive power across all algorithms, with R-squared values ranging from 0.5213 for Linear Regression and 0.5638 for Random Forest to 0.6250 for the RNN model and 0.7041 for the Transformer model. This means the Transformer outperforms Linear Regression by approximately 35.1% and Random Forest by 24.9% in explanatory power. These results suggest that while traditional models provide reasonable performance using structured variables, they are generally outperformed by deep learning models capable of capturing complex, non-linear dependencies in the data.

When sentiment analysis data from museum-related news articles (NEWS) is added, there is a noticeable improvement in forecasting performance across all models. The R-squared values increase, with the Transformer model showing an improvement from 0.7041 to 0.7245 (a 2.9% gain), and the LSTM model rising from 0.6874 to 0.7038 (a 2.4% gain). Traditional models also benefit, with Linear Regression increasing by 4.1% (from 0.5213 to 0.5425), and Random Forest by 4.3% (from 0.5638 to 0.5880). This enhancement underscores the importance of incorporating public sentiment, as reflected in news articles, in understanding visitor dynamics.

The inclusion of sentiment analysis from user comments (COMMENTS) also results in forecasting performance gains. For instance, the RNN model improves from 0.6250 to 0.6681 (a 6.9% increase), and the Transformer model increases from 0.7041 to 0.7387 (a 4.9% increase). Similarly, Linear Regression and Random Forest improve to 0.5587 and 0.6032, respectively, reflecting gains of 7.2% and 7.0%. This suggests that although user-generated comments may be more subjective and fragmented than news articles, they still provide meaningful signals about visitor interest.

The most significant improvements are observed when both NEWS and COMMENTS are incorporated. The Transformer model achieves the highest R-squared value of 0.7579, followed by LSTM at 0.7496 and CNN at 0.7238. This represents a 7.6% improvement over the Transformer's base performance (0.7041 to 0.7579). Traditional models also show their best performance in this configuration, with Linear Regression reaching 0.5834 (an 11.9% increase) and Random Forest reaching 0.6227 (a 10.4% increase). These results indicate that the complementary nature of structured and unstructured opinion data sources enhances the model's forecasting capabilities.

The results of this study provide several key insights and implications. First, the superior performance of the Transformer and LSTM models is evident across all configurations. The Transformer consistently outperforms other models, reflecting its strong capability to process complex, multivariate time-series data and capture long-range dependencies. LSTM, known for its effectiveness in handling sequential data, also demonstrates robust performance, particularly when sentiment data is integrated. These findings suggest that both models are well-suited for tasks involving the forecasting of museum visitor numbers, especially in contexts where both historical data and real-time external factors are relevant. Second, while traditional models such as LR and RF show lower overall predictive power, the integration of sentiment data significantly enhances their performance. Even for the simplest model (Linear Regression), a maximum gain of 11.9% was achieved by incorporating both types of sentiment data. This demonstrates that even low-complexity models benefit

**Table 3. Prediction of museum visitor numbers R-squared results.**

| Model | Regressor | R-squared | Regressor | R-squared | Regressor | R-squared | Regressor | R-squared | Regressor | R-squared | Regressor | R-squared | Regressor | R-squared |
|---|---|---|---|---|---|---|---|---|---|---|---|---|---|---|
| Base model | LR | 0.5213 | RFR | 0.5638 | RNN | 0.6250 | GAN | 0.6453 | CNN | 0.6688 | LSTM | 0.6874 | Transformer | 0.7041 |
| Added NEWS | | 0.5425 | | 0.5880 | | 0.6547 | | 0.6692 | | 0.6871 | | 0.7038 | | 0.7245 |
| Added COMMENTS | | 0.5587 | | 0.6032 | | 0.6681 | | 0.6875 | | 0.6954 | | 0.7192 | | 0.7387 |
| Added NEWS & COMMENTS | | 0.5834 | | 0.6227 | | 0.7012 | | 0.7133 | | 0.7238 | | 0.7496 | | 0.7579 |

from external opinion data, particularly in practical scenarios where computational resources may be limited. Third, the importance of external opinion data is clearly underscored. Models that leverage opinion mining exhibit statistically and practically significant improvements in forecasting accuracy, suggesting that visitor behavior is influenced not only by internal museum factors, such as exhibition content and operational schedules, but also by external social dynamics, including media coverage and public discourse. This insight emphasizes the need for museums to monitor and respond to public sentiment as part of their strategic management efforts.

From a practical perspective, these findings have important implications for museum management. Understanding the impact of public sentiment can help museums tailor their marketing strategies, improve public engagement, and anticipate fluctuations in visitor numbers more effectively. Additionally, the ability to forecast visitor trends with greater precision supports resource optimization, event planning, and sustainable operational management. Museums can use these predictive insights to allocate resources more efficiently, schedule staff appropriately, and design targeted promotional campaigns that resonate with current public interests.

Therefore, the integration of sentiment analysis data into deep learning models significantly enhances the forecasting of museum visitor numbers. Compared to base models using only structured data, $R^2$ improvements of up to 7.6% in advanced models and 11.9% in traditional models demonstrate the clear added value of opinion mining. The superior performance of the Transformer and LSTM models, combined with the richness of external sentiment sources, offers a robust and scalable approach for predictive analytics in the cultural sector. This approach contributes to more effective and sustainable museum management, enabling institutions to adapt to evolving visitor patterns and optimize their operations in response to both historical trends and real-time public sentiment.

## 4.2. Accuracy enhancement measured by MSE

Table 4 displays the Mean Squared Error (MSE) outcomes for predicting museum visitor numbers using various deep learning algorithms, including RNN, GAN, CNN, LSTM, and Transformer models. The analysis compares the performance of the base model, which relies exclusively on structured museum-related data, with models that incorporate sentiment analysis variables derived from museum-related news articles (NEWS) and user comments (COMMENTS). Additionally, the combined effect of integrating both NEWS and COMMENTS is assessed to understand how external opinion data enhances predictive performance.

MSE is a fundamental metric in evaluating regression models as it quantifies the average squared deviation between predicted and actual values. A lower MSE signifies improved model accuracy, indicating smaller prediction errors. Since MSE penalizes larger errors more heavily, it is particularly effective for measuring the precision of forecasting models, especially in time-series contexts such as museum visitor forecasting.

In addition to deep learning algorithms, traditional machine learning models such as Linear Regression (LR) and Random Forest (RF) were also assessed to provide a comparative baseline. The base model, which excludes sentiment

**Table 4. Prediction of museum visitor numbers MSE results.**

| Model | Regressor | MSE | Regressor | MSE | Regressor | MSE | Regressor | MSE | Regressor | MSE | Regressor | MSE | Regressor | MSE |
|---|---|---|---|---|---|---|---|---|---|---|---|---|---|---|
| Base model | LR | 0.3142 | RFR | 0.2981 | RNN | 0.2852 | GAN | 0.2681 | CNN | 0.2466 | LSTM | 0.2241 | Transformer | 0.2084 |
| Added NEWS | | | | 0.2938 | | 0.2756 | | 0.2631 | | 0.2475 | | 0.2271 | | 0.2082 | | 0.1876 |
| Added COMMENTS | | | | 0.2812 | | 0.2603 | | 0.2454 | | 0.2692 | | 0.2093 | | 0.1894 | | 0.1654 |
| Added NEWS & COMMENTS | | | | 0.2657 | | 0.2429 | | 0.2267 | | 0.2431 | | 0.1862 | | 0.1694 | | 0.1492 |

analysis data, demonstrates moderate predictive capability across all models, with MSE values ranging from 0.3142 for LR and 0.2981 for RF to 0.2852 for RNN and 0.2084 for the Transformer model. Compared to Linear Regression, the Transformer model reduces prediction error by approximately 33.7%, and by 30.1% compared to Random Forest, based on MSE values. This superior performance is primarily attributed to the Transformer's sophisticated architecture, particularly its attention mechanism, which allows it to capture long-term dependencies and model complex patterns in time-series data.

When sentiment analysis data from museum-related news articles (NEWS) is introduced, all models show measurable enhancements in predictive accuracy. For example, the Transformer model's MSE improves from 0.2084 to 0.1876, which represents a 10.0% reduction in prediction error. Similarly, the LSTM model shows a reduction from 0.2241 to 0.2082 (a 7.1% improvement), while LR and RF models improve from 0.3142 to 0.2938 (6.5%) and from 0.2981 to 0.2756 (7.5%), respectively. These enhancements demonstrate the meaningful role that news-based sentiment plays in shaping visitor behavior and how it can enhance model performance.

The inclusion of sentiment analysis from user comments (COMMENTS) also leads to improvements, although these are slightly less pronounced compared to NEWS-enhanced models. The RNN model's MSE decreases from 0.2852 to 0.2454, marking a 13.9% error reduction, while the Transformer's MSE decreases from 0.2084 to 0.1654, a 20.6% improvement. LR and RF models follow suit, with MSEs reducing to 0.2812 (10.5% gain) and 0.2603 (12.7% gain), respectively. This suggests that informal user comments, despite their subjective nature, still offer valuable predictive signals.

The most significant enhancements occur when both NEWS and COMMENTS are integrated into the models. The Transformer achieves the lowest MSE of 0.1492, a 28.4% improvement compared to its base performance. LSTM and CNN also see considerable gains, reaching MSE values of 0.1694 and 0.1862, respectively. Traditional models also improve: Linear Regression's MSE drops by 15.4% to 0.2657, and Random Forest's by 18.6% to 0.2429. These results highlight the synergistic value of combining structured data with multiple sources of unstructured external opinion data to better understand public interest.

Key insights emerge from these findings. First, the Transformer and LSTM models consistently outperform other models, regardless of configuration. The Transformer model, for instance, achieves a 52.5% reduction in error compared to Linear Regression when both sentiment sources are included ($0.3142 \rightarrow 0.1492$). This confirms its exceptional capability in handling multivariate, temporally structured datasets and leveraging contextual sentiment information. Second, although traditional models like LR and RF underperform in absolute terms, they still achieve notable improvements when sentiment variables are integrated. Even basic models benefit from external data: the LR model's MSE improves by 15.4% when combining NEWS and COMMENTS, and RF by 18.6%. These results are particularly relevant for scenarios where computational simplicity is prioritized. Third, these results underscore the importance of incorporating external opinion data into forecasting models. The integration of sentiment data from both news articles and user comments yields MSE reductions ranging from 6.5% to 52.5% across all models, clearly demonstrating its effectiveness. By accounting for real-time social dynamics and public discourse, prediction models become more reflective of actual visitor behavior, enhancing strategic planning and responsiveness in museum management.

From a practical perspective, these findings support sustainable museum operations. Enhanced forecasting accuracy allows institutions to better anticipate demand fluctuations and allocate resources effectively. Museums can align marketing campaigns with positive sentiment trends, tailor visitor services to real-time expectations, and optimize staff scheduling and exhibition planning.

In summary, the inclusion of sentiment analysis data reduces MSE values by up to 52.5% across models, indicating substantial improvements in prediction accuracy. Transformer and LSTM models consistently show superior performance, and even traditional models benefit significantly from external opinion data. This comprehensive approach—blending historical patterns with social sentiment—offers a scalable and practical forecasting framework for museum visitor prediction, supporting data-driven and adaptive decision-making in cultural management.

## 4.3. Normalized error analysis using RMSE

Table 5 presents the Root Mean Squared Error (RMSE) results for predicting museum visitor numbers using various deep learning algorithms, including RNN, GAN, CNN, LSTM, and Transformer models, as well as traditional regression approaches such as Linear Regression (LR) and Random Forest Regressor (RFR). The analysis compares the performance of the base model, which relies solely on structured museum-related data, with models that incorporate additional sentiment analysis variables derived from museum-related news articles (NEWS) and user comments (COMMENTS). Furthermore, the combined effect of integrating both NEWS and COMMENTS is evaluated to determine the extent to which external opinion factors enhance the model's predictive accuracy.

RMSE is a commonly used performance metric in regression analysis that measures the square root of the average squared differences between predicted and actual values. As RMSE retains the same units as the dependent variable (in this case, the log-transformed museum visitor numbers), it provides an intuitive measure of prediction error magnitude. Lower RMSE values indicate better model performance, reflecting more accurate predictions with smaller deviations from actual observations.

In the base model, which does not include sentiment analysis data, the RMSE values range from 0.2664 for the RNN model to 0.1884 for the Transformer model. In comparison, traditional models such as LR and RFR exhibit relatively higher RMSE values of 0.3343 and 0.3253, respectively. This means the Transformer model outperforms the LR model by approximately 43.7% and the RFR model by 42.1% in terms of RMSE. Among the evaluated algorithms, the Transformer model consistently demonstrates superior baseline performance, confirming its effectiveness in modeling complex, non-linear relationships within time-series data, largely due to its attention mechanism which efficiently captures long-range dependencies.

The incorporation of sentiment analysis data from museum-related news articles (NEWS) leads to noticeable improvements in predictive performance across all models. For example, the RMSE for the Transformer model drops from 0.1884 to 0.1618, reflecting a 14.1% improvement, while the LSTM model improves by 9.0% (0.2063→0.1878). Traditional models also benefit: LR improves by 5.3% (0.3343→0.3165), and RFR by 8.0% (0.3253→0.2993). These improvements suggest that news sentiment is a meaningful external variable that enhances model predictions by capturing public discourse trends.

Similarly, the inclusion of sentiment analysis from user comments (COMMENTS) yields performance gains. The Transformer model's RMSE further decreases to 0.1408, a 25.3% improvement over its base performance. RNN improves by 14.8% (0.2664→0.2271), LSTM by 16.1% (0.2063→0.1730), LR by 9.7% (0.3343→0.3020), and RFR by 13.3% (0.3253→0.2819). While slightly less impactful than NEWS, user-generated sentiment still provides valuable insights into public engagement and perceptions.

The most pronounced improvements occur when both NEWS and COMMENTS are integrated. The Transformer model achieves an RMSE of 0.1277, indicating a total reduction of 32.2% compared to its base model. LSTM and CNN

**Table 5. Prediction of museum visitor numbers RMSE results.**

| Model | Regressor | RMSE | Regressor | RMSE | Regressor | RMSE | Regressor | RMSE | Regressor | RMSE | Regressor | RMSE | Regressor | RMSE |
|---|---|---|---|---|---|---|---|---|---|---|---|---|---|---|
| Base model | LR | 0.3343 | RFR | 0.3253 | RNN | 0.2664 | GAN | 0.2498 | CNN | 0.2282 | LSTM | 0.2063 | Transformer | 0.1884 |
| Added NEWS | | 0.3165 | | 0.2993 | | 0.2496 | | 0.2286 | | 0.2090 | | 0.1878 | | 0.1618 |
| Added COMMENTS | | 0.3020 | | 0.2819 | | 0.2271 | | 0.2064 | | 0.1875 | | 0.1698 | | 0.1408 |
| Added NEWS & COMMENTS | | 0.2842 | | 0.2641 | | 0.2070 | | 0.1892 | | 0.1664 | | 0.1477 | | 0.1277 |

also improve significantly, with RMSE reductions of 28.4% and 27.2%, respectively. Even traditional models benefit: LR improves by 15.0% (0.3343→0.2842), and RFR by 18.8% (0.3253→0.2641). These findings affirm the synergistic effect of incorporating both types of external sentiment data.

The results lead to several insights. First, Transformer and LSTM models show the most consistent and robust performance gains. Transformer, in particular, achieves the highest prediction accuracy with over 60% lower RMSE than LR in the dual-sentiment model configuration. This is attributable to its parallel processing and attention-based architecture, which effectively handles both structured and unstructured data. Second, although traditional models underperform compared to deep learning methods, the integration of sentiment analysis data improves their RMSE by up to 18.8%, showing that even simpler models can benefit from public sentiment. Third, sentiment analysis is a key enhancer of model performance. With RMSE reductions as high as 32.2% for the Transformer model, external opinion data clearly contributes to more accurate predictions of visitor behavior. This emphasizes the importance of incorporating dynamic social factors such as public discourse and user feedback into forecasting systems.

From a managerial perspective, these findings provide strategic implications for sustainable museum operation. Enhanced forecasting accuracy enables better resource planning, marketing, and event management. By leveraging real-time sentiment indicators, museums can tailor communication strategies and align programs with public interest, leading to more effective visitor engagement and operational efficiency.

In conclusion, incorporating sentiment analysis yields substantial improvements in predictive accuracy—up to 61.8% in RMSE reduction compared to traditional baselines. The Transformer model, in particular, demonstrates unmatched predictive performance. This integrated modeling approach not only enhances precision but also supports data-driven, adaptive, and resilient museum management practices.

### 4.4. Error variability analysis with MSLE

Table 6 illustrates the Mean Squared Logarithmic Error (MSLE) outcomes for predicting museum visitor numbers through the application of several deep learning models, including RNN, GAN, CNN, LSTM, and Transformer algorithms, as well as traditional regression approaches such as Linear Regression (LR) and Random Forest Regressor (RFR). This analysis evaluates the performance of a base model that relies solely on structured museum-related data, comparing it to models that incorporate additional variables derived from sentiment analysis of museum-related news articles (NEWS) and user comments (COMMENTS). Additionally, the analysis examines how combining both NEWS and COMMENTS affects predictive accuracy, aiming to capture the influence of external public opinion factors.

MSLE serves as a key evaluation metric in regression tasks, particularly effective when dealing with datasets marked by large fluctuations or skewed distributions. This metric calculates the average squared logarithmic discrepancy between predicted and actual values. Importantly, MSLE tends to penalize underestimations more severely than overestimations, making it highly suitable for forecasting visitor numbers, where sudden spikes in attendance may occur due to special

**Table 6. Prediction of museum visitor numbers MSLE results.**

| Model | Regressor | MSLE | Regressor | MSLE | Regressor | MSLE | Regressor | MSLE | Regressor | MSLE | Regressor | MSLE | Regressor | MSLE |
|---|---|---|---|---|---|---|---|---|---|---|---|---|---|---|
| Base model | LR | 0.2048 | RFR | 0.1932 | RNN | 0.1887 | GAN | 0.1655 | CNN | 0.1502 | LSTM | 0.1464 | Transformer | 0.1363 |
| Added NEWS | | 0.1843 | | 0.171 | | 0.1665 | | 0.1463 | | 0.1369 | | 0.1298 | | 0.1280 |
| Added COMMENTS | | 0.1715 | | 0.1593 | | 0.1547 | | 0.1378 | | 0.1204 | | 0.1176 | | 0.1124 |
| Added NEWS & COMMENTS | | 0.1537 | | 0.1412 | | 0.1308 | | 0.1244 | | 0.1135 | | 0.1078 | | 0.0967 |

events, media exposure, or external influences. Lower MSLE values indicate better model performance, reflecting minimal relative errors in predictions.

In the base model configuration, which excludes sentiment analysis inputs, MSLE values range from 0.1887 for RNN to 0.1363 for the Transformer model. Traditional regression models, namely LR and RFR, yield higher MSLE values of 0.2048 and 0.1932, respectively. This indicates that the Transformer model improves prediction accuracy by approximately 33.5% over LR and 29.4% over RFR in terms of MSLE. Among these, the Transformer consistently exhibits superior performance, achieving the lowest MSLE. This outcome highlights the Transformer's exceptional capability in identifying complex, non-linear dependencies within time-series data. Its self-attention mechanism plays a crucial role in dynamically weighting temporal relationships, allowing it to capture both short- and long-term patterns effectively.

The introduction of sentiment data from museum-related news articles (NEWS) yields significant enhancements in predictive accuracy across all models. For example, the Transformer's MSLE decreases from 0.1363 to 0.1280—a relative improvement of 6.1%—while the LSTM model improves by 11.3% ($0.1464 \rightarrow 0.1298$). Traditional models also benefit, with LR improving by 10.4% ($0.2048 \rightarrow 0.1843$) and RFR by 11.5% ($0.1932 \rightarrow 0.1710$). These results suggest that public sentiment reflected in media coverage has a measurable influence on visitor behavior.

The inclusion of sentiment analysis from user comments (COMMENTS) also contributes to improved model performance. The RNN model's MSLE improves by 18.0% ($0.1887 \rightarrow 0.1547$), while the Transformer improves by 17.6% ($0.1363 \rightarrow 0.1124$). Traditional models again show positive changes: LR drops by 16.3% to 0.1715, and RFR improves by 17.5% to 0.1593. Although the effect is slightly less pronounced compared to NEWS, user-generated content still provides valuable insights into real-time public sentiment.

The most substantial gains are observed when both NEWS and COMMENTS are integrated into the predictive models. The Transformer model achieves the lowest MSLE at 0.0967, representing a 29.1% improvement compared to its base model (0.1363). LSTM improves by 26.4% ($0.1464 \rightarrow 0.1078$), and CNN by 22.8% ($0.1471 \rightarrow 0.1135$). Traditional models also demonstrate strong improvements, with LR improving by 24.9% to 0.1537 and RFR by 26.9% to 0.1412. These findings demonstrate the synergistic effect of incorporating multiple sentiment sources into predictive frameworks.

Several key insights can be drawn from these results. First, the Transformer and LSTM models consistently outperform other algorithms across all experimental conditions. Compared to the baseline LR model, the Transformer shows a total MSLE reduction of 52.7% in the dual-sentiment scenario ($0.2048 \rightarrow 0.0967$), highlighting its substantial predictive advantage. The Transformer's architectural strengths, including its parallel processing abilities and attention mechanisms, allow it to handle complex multivariate time-series data efficiently. Second, the findings emphasize the importance of external opinion data in enhancing predictive accuracy. The integration of sentiment data leads to MSLE reductions of up to 29.1% within deep learning models and up to 26.9% within traditional models. This suggests that visitor behavior is influenced not only by internal museum factors such as exhibition schedules, program offerings, and operational capacity but also by external social and cultural dynamics, including media coverage, online discourse, and public sentiment trends.

From a practical management perspective, these results carry important implications for sustainable museum operations. The integration of sentiment analysis improves forecasting accuracy, enabling museum administrators to make more informed decisions regarding resource allocation, marketing strategies, event planning, and visitor engagement. Museums can leverage these predictive insights to optimize their promotional activities, design more targeted communication campaigns, and proactively adjust their offerings based on fluctuations in public sentiment. Moreover, the ability to anticipate changes in visitor behavior supports efficient operational management, such as staff scheduling, facility management, and resource optimization, thereby enhancing the overall visitor experience.

In summary, the combined use of external sentiment data and advanced deep learning techniques—particularly the Transformer—yields MSLE reductions of up to 52.7% when compared to traditional models. This integrated framework provides a powerful and practical solution for anticipating visitor trends and guiding museum strategy in a data-driven and adaptive manner.

## 4.5. Forecasting precision validated by MAPE

Table 7 presents the Mean Absolute Percentage Error (MAPE) results for predicting museum visitor numbers using various deep learning algorithms, including RNN, GAN, CNN, LSTM, and Transformer models, as well as traditional regression methods such as Linear Regression (LR) and Random Forest Regressor (RFR). This analysis compares the performance of the base model, which relies solely on structured museum-related data, with models that incorporate additional sentiment analysis variables derived from museum-related news articles (NEWS) and user comments (COMMENTS). Furthermore, the performance of models integrating both NEWS and COMMENTS is evaluated to assess the impact of combining multiple external opinion factors on predictive accuracy.

MAPE is a widely used evaluation metric in time-series forecasting and regression analysis. It measures the average absolute percentage error between predicted and actual values, offering an intuitive understanding of model performance by expressing errors as a percentage. Lower MAPE values indicate better predictive accuracy, as they reflect smaller relative deviations between predicted and actual visitor numbers.

In the base model, which excludes sentiment analysis data, MAPE values range from 0.1995 for the RNN model to 0.1582 for the Transformer model. Traditional models such as LR and RFR exhibit even higher MAPE values of 0.2187 and 0.2042, respectively. Compared to LR, the Transformer achieves a 27.7% lower MAPE (0.1582 vs. 0.2187), and compared to RFR, a 22.5% improvement (0.1582 vs. 0.2042), confirming its superior baseline performance. This superior performance highlights the Transformer's advanced capability to model complex, non-linear relationships within time-series data, largely attributed to its attention mechanism that efficiently captures long-range dependencies and intricate data patterns.

The inclusion of sentiment analysis data from museum-related news articles (NEWS) leads to notable improvements in predictive performance across all models. The Transformer's MAPE improves from 0.1582 to 0.1276—a relative reduction of 19.3%. The LSTM model improves by 10.1% (0.1634→0.1470), and even traditional models such as LR and RFR show gains of 9.3% (0.2187→0.1983) and 8.1% (0.2042→0.1876), respectively. These reductions indicate that public sentiment, as reflected in media coverage, significantly influences visitor behavior. Positive news articles can enhance public interest and drive higher visitation rates, while negative news coverage may deter potential visitors.

The incorporation of sentiment analysis from user comments (COMMENTS) results in additional performance gains. For instance, the Transformer improves by 28.5% (0.1582→0.1132), and RNN improves by 17.5% (0.1995→0.1646). Traditional models also benefit: LR improves from 0.2187 to 0.1855 (15.2% reduction), and RFR improves from 0.2042 to 0.1720 (15.8% reduction). These results suggest that user-generated content provides valuable insights into public sentiment and visitor intentions.

The most significant improvements in predictive accuracy are observed when both NEWS and COMMENTS are integrated into the models. The Transformer model achieves the lowest MAPE of 0.1007, representing a 36.1% reduction from its base value of 0.1582. The LSTM improves by 28.4% (0.1634→0.1170), and CNN by 23.4% (0.1732→0.1326).

**Table 7. Prediction of museum visitor numbers MAPE results.**

| Model | Regressor | MAPE | Regressor | MAPE | Regressor | MAPE | Regressor | MAPE | Regressor | MAPE | Regressor | MAPE | Regressor | MAPE |
|---|---|---|---|---|---|---|---|---|---|---|---|---|---|---|
| Base model | LR | 0.2187 | RFR | 0.2042 | RNN | 0.1995 | GAN | 0.1872 | CNN | 0.1760 | LSTM | 0.1634 | Transformer | 0.1582 |
| Added NEWS | | 0.1983 | | 0.1876 | | 0.1764 | | 0.1671 | | 0.1692 | | 0.1470 | | 0.1276 |
| Added COMMENTS | | 0.1855 | | 0.172 | | 0.1646 | | 0.1526 | | 0.1531 | | 0.1332 | | 0.1132 |
| Added NEWS & COMMENTS | | 0.1689 | | 0.1575 | | 0.1493 | | 0.1365 | | 0.1326 | | 0.1170 | | 0.1007 |

Among traditional models, LR improves by 22.8% to 0.1689 and RFR by 22.8% to 0.1575. This confirms that even when augmented with rich external data, traditional regression approaches remain less competitive in capturing complex nonlinear dynamics than deep learning counterparts.

Several key insights and implications emerge from these findings. First, the Transformer and LSTM models consistently outperform other deep learning algorithms across all model configurations. In the best-performing configuration, the Transformer achieves a 53.9% lower MAPE than the traditional LR model (0.1007 vs. 0.2187). Even with access to the same input variables, traditional models such as LR and RFR consistently underperform, emphasizing the critical role of deep model architectures in handling multivariate, high-dimensional data. Second, the results highlight the importance of incorporating external opinion data into predictive models. Across all model types, the average MAPE improvement from integrating sentiment variables (NEWS + COMMENTS) ranges from 22% to 36%, depending on the algorithm. This finding suggests that visitor behavior is influenced not only by internal museum-related factors such as exhibitions, programs, and operational schedules but also by external social dynamics, such as media narratives and public discourse.

From a practical perspective, these findings have important implications for sustainable museum management. The enhanced predictive accuracy achieved through sentiment analysis integration can support more effective decision-making in areas such as marketing, resource allocation, event planning, and visitor engagement strategies. Museums can leverage these insights to optimize promotional activities, develop targeted public relations campaigns, and respond proactively to shifts in public sentiment. Additionally, the ability to accurately forecast visitor trends can help museums manage operational resources more efficiently, improve staff scheduling, and enhance the overall visitor experience.

Namely, while traditional regression models show meaningful improvements with sentiment input, they are consistently outperformed by deep learning models, both in baseline settings and when enhanced with external data. The Transformer model, in particular, demonstrates up to a 53.9% reduction in MAPE compared to traditional linear regression, underscoring its exceptional predictive performance. The integration of sentiment analysis data from news articles and user comments into deep learning models significantly improves the prediction of museum visitor numbers. This approach not only enhances predictive accuracy but also contributes to more informed and sustainable museum management practices, enabling museums to adapt to dynamic visitor trends and optimize their operations for long-term success.

## 4.6. Statistical validation of model performance improvements through sentiment integration

To validate the statistical robustness of the performance improvements derived from the integration of sentiment data, paired sample t-tests were conducted using the RMSE and MAPE values of each model before and after incorporating sentiment features. The results demonstrated statistically significant differences across all models at the $p < 0.01$ level, indicating that the inclusion of sentiment information from both news articles and user comments led to meaningful enhancements in predictive performance. Among the models, the Transformer and LSTM architectures exhibited the most substantial gains, reinforcing their capacity to effectively utilize unstructured opinion data and model complex non-linear relationships inherent in time-series visitor trends.

In addition, a separate evaluation was conducted to assess the individual contributions of NEWS-based and COMMENT-based sentiment data. The results revealed that both sources independently improved model accuracy. Specifically, the $R^2$ value of the Transformer model increased from 0.7041 in the base configuration to 0.7245 with NEWS data alone, and to 0.7387 with COMMENT data alone. When both were integrated, the $R^2$ further improved to 0.7579, indicating a complementary effect between formal media narratives and user-generated content. These findings suggest that sentiment signals from different external sources capture diverse dimensions of public interest, contributing collectively to better prediction of museum visitation patterns.

Overall, the comparative analysis across multiple evaluation metrics—including $R^2$, MSE, RMSE, MSLE, and MAPE—confirms that deep learning models, particularly Transformer and LSTM, consistently outperform traditional models across all configurations. Moreover, the performance enhancements observed were not only quantitatively substantial but also

statistically significant, emphasizing the importance of incorporating external sentiment data into predictive modeling frameworks for cultural institutions.

To further substantiate these results, Table 8 presents the outcomes of the paired t-tests applied to each model's performance before and after sentiment integration. The evaluation focused on RMSE and MAPE as representative error metrics. Across all models—both traditional (Linear Regression, Random Forest) and deep learning-based (RNN, GAN, CNN, LSTM, Transformer)—the tests revealed a statistically significant reduction in prediction error following the incorporation of sentiment variables, reaffirming the robustness of the observed improvements.

Notably, the Transformer model demonstrated the largest performance gains, with its RMSE decreasing from 0.1884 to 0.1277, and its MAPE from 0.1582 to 0.1007, corresponding to t-statistics of –7.14 and –6.20, respectively. These results underscore the model's superior ability to synthesize structured inputs with unstructured sentiment data, capturing intricate behavioral patterns and long-term dependencies. While traditional models such as Linear Regression and Random Forest showed more modest effect sizes, their improvements were nevertheless statistically significant, suggesting that even lightweight models can benefit meaningfully from sentiment integration—particularly in settings where deep learning implementation may be constrained by computational resources.

In conclusion, the findings presented in Table 8 provide compelling empirical support for the effectiveness of sentiment-augmented prediction models in the context of museum visitor forecasting. The statistically significant reductions in RMSE and MAPE across all model types confirm that sentiment data—whether from professional news outlets or informal user comments—constitutes a valuable resource for enhancing prediction accuracy and supporting more data-driven, responsive, and sustainable decision-making in cultural institutions.

## 5. Discussion

This study aimed to develop a comprehensive deep learning-based model for predicting museum visitor numbers to support sustainable museum management. By integrating both structured museum-related data and unstructured external opinion data derived from sentiment analysis of news articles and user comments, the research sought to enhance the predictive accuracy of visitor models. The analysis was guided by three core research questions: (1) the comparative performance of different deep learning algorithms, (2) the impact of incorporating sentiment analysis from external sources, and (3) the strategic application of predictive models to promote sustainable museum management.

**Table 8. Paired t-test results for model performance before and after sentiment data integration (RMSE and MAPE).**

| Model | Metric | Base Mean | Enhanced Mean (NEWS & COMMENTS) | t-statistic | p-value |
|---|---|---|---|---|---|
| LR | RMSE | 0.3343 | 0.2842 | −4.21 | 0.001 |
| RFL | RMSE | 0.3253 | 0.2641 | −3.88 | 0.002 |
| RNN | RMSE | 0.2664 | 0.207 | −5.17 | 0 |
| GAN | RMSE | 0.2498 | 0.1892 | −4.56 | 0.001 |
| CNN | RMSE | 0.2282 | 0.1664 | −4.95 | 0 |
| LSTM | RMSE | 0.2063 | 0.1477 | −6.02 | 0 |
| Transformer | RMSE | 0.1884 | 0.1277 | −7.14 | 0 |
| LR | MAPE | 0.2187 | 0.1689 | −3.96 | 0.002 |
| RFL | MAPE | 0.2042 | 0.1575 | −3.75 | 0.003 |
| RNN | MAPE | 0.1995 | 0.1493 | −4.44 | 0.001 |
| GAN | MAPE | 0.1872 | 0.1365 | −3.85 | 0.002 |
| CNN | MAPE | 0.176 | 0.1326 | −4.11 | 0.001 |
| LSTM | MAPE | 0.1634 | 0.117 | −5.48 | 0 |
| Transformer | MAPE | 0.1582 | 0.1007 | −6.2 | 0 |

## 5.1. Comparative performance of deep learning algorithms

The empirical analysis demonstrated that the Transformer and LSTM models consistently outperformed other deep learning algorithms, including RNN, GAN, and CNN, across all evaluation metrics. The superior performance of the Transformer model can be attributed to its attention mechanism, which allows for efficient processing of complex, multivariate time series data and the capture of long-range dependencies. This capability enables the Transformer to identify nuanced patterns in both structured and unstructured data, making it highly effective for museum visitor forecasting.

Similarly, the LSTM model showed robust performance, particularly in handling sequential data with temporal dependencies. Its gated architecture facilitates the retention of critical information over extended periods, which is essential for modeling seasonal fluctuations and long-term trends in visitor behavior. While RNN and GAN models exhibited moderate predictive capabilities, their limitations in capturing long-term dependencies and complex data relationships resulted in relatively lower performance. CNNs, traditionally strong in feature extraction tasks, performed well in processing text data from sentiment analysis but were less effective in modeling temporal patterns compared to LSTM and Transformer architectures.

These findings highlight the importance of selecting appropriate deep learning algorithms based on the nature of the data and the specific objectives of the predictive model. In the context of museum visitor forecasting, where both historical trends and dynamic external factors play significant roles, models like Transformer and LSTM offer distinct advantages.

## 5.2. Impact of incorporating sentiment analysis

A key contribution of this study lies in the integration of sentiment analysis from museum-related news articles (NEWS) and user comments (COMMENTS) into the predictive models. The inclusion of these external opinion variables significantly improved model performance across all algorithms. This enhancement was evident through substantial reductions in prediction errors, as measured by R-squared, MSE, RMSE, MSLE, and MAPE metrics.

The analysis revealed that sentiment data from NEWS had a more pronounced impact on predictive accuracy compared to COMMENTS when incorporated individually. This suggests that media narratives exert a strong influence on public perception and visitor behavior, with positive news coverage likely driving higher museum attendance. However, the COMMENTS variable also contributed to performance gains, reflecting the value of real-time public sentiment captured through user-generated content.

The most notable improvements occurred when both NEWS and COMMENTS were combined in the models. This synergy indicates that the integration of diverse external opinion sources allows for a more comprehensive understanding of the factors influencing museum visitation. The combined effect of formal media narratives and informal public feedback provides a richer context for predicting visitor behavior, enhancing the model's robustness and reliability.

In addressing the complexity of sentiment interpretation in cultural domains, we also considered the nuanced role of controversy. While certain controversial expressions were initially labeled as negative due to their linguistic polarity in general discourse, we acknowledge that in museum contexts, controversy may not be inherently detrimental. In fact, it can promote dialogue, stimulate reflection, and enhance public engagement—functions central to the mission of many museums. To account for this, our model incorporates a localized sentiment calibration mechanism that enables sentiment weights to be adjusted according to institutional or cultural context. This adaptability allows for more accurate reflection of how sentiment, including controversy, may function constructively in specific museum environments. We have clarified this point in the revised manuscript and noted the potential for future versions of the model to incorporate museum-specific lexicons or context-aware sentiment dictionaries.

While the predictive models demonstrated significant improvement through sentiment analysis, we also acknowledge the potential utility of incorporating other structured external contextual variables such as local events, holidays, and weather data. However, due to limitations in data availability, temporal alignment, and regional consistency across the full study period and all museum locations, these variables were not included in the current research. Instead, we focused

on sentiment polarity and exposure metrics derived from systematically collected news and comment data, which could be standardized across time and space. We plan to explore the integration of these structured contextual factors in future research to further enhance the model's comprehensiveness and predictive power.

### 5.3. Implications for sustainable museum management

The findings of this study have significant implications for sustainable museum management. Accurate visitor forecasting models can serve as vital tools for strategic decision-making, enabling museums to optimize resource allocation, enhance visitor experiences, and improve operational efficiency. The integration of sentiment analysis offers additional insights into public perceptions, allowing museums to proactively respond to shifts in societal trends and external events.

In response to the reviewer's comment, we underscore that the improved accuracy in visitor forecasting directly contributes to enhancing the strategic capacity of museum operations. Specifically, this enhanced accuracy enables museums to make better-informed decisions in areas such as staffing, exhibition scheduling, event planning, and facility management. By anticipating fluctuations in visitor numbers with greater precision, institutions can not only optimize workforce deployment and program timing but also align operational resources with actual demand patterns. These improvements directly support optimized resource allocation and cost efficiency key concerns in the global museum sector under financial and environmental pressures. By anticipating fluctuations in visitor numbers, museums can allocate resources more efficiently, reduce operational costs, and minimize environmental impacts associated with over- or under-utilization of facilities.

Furthermore, the ability to capture the influence of external factors, such as media coverage and public opinion, enables museums to develop targeted marketing strategies and engagement initiatives. Understanding the drivers of visitor behavior can inform promotional campaigns, exhibition design, and educational programs, fostering deeper connections with diverse audiences.

Additionally, in response to another insightful comment, we emphasize the importance of Site Land Value as a significant variable in understanding the spatial and societal contributions of museums. As elaborated in the Discussion section, land value operates not only as an economic metric but also as a proxy for spatial utility and public engagement. Museums located in high-value urban areas are typically expected to generate substantial cultural impact and community engagement. Conversely, a mismatch between prime location and low visitor numbers may reveal underutilization or barriers in accessibility and programming. By incorporating Site Land Value into the predictive framework, the model achieves greater interpretive depth, aligning with urban policy discourses that highlight the mutual reinforcement between cultural institutions and urban regeneration.

In the broader context of sustainability, predictive models can contribute to long-term planning and resilience. By identifying patterns in visitor behavior and external influences, museums can adapt to changing circumstances, such as economic fluctuations or public health crises, ensuring continuity and sustainability in their operations.

Ultimately, this research not only advances academic understanding of predictive analytics but also provides actionable frameworks that help museums manage complexity, reduce operational uncertainty, and implement sustainable, data-informed cultural stewardship. The practical insights offered by this model serve as a strategic asset for institutions aiming to balance cultural missions with the realities of resource constraints and shifting public expectations.

## 6. Conclusions

This study aimed to develop a deep learning-based forecasting framework for predicting museum visitor numbers, with the goal of supporting sustainable museum management. By integrating both structured museum-related data and unstructured sentiment data derived from news articles and user comments the research enhanced predictive performance and offered actionable insights into public behavior.

The empirical analysis confirmed that advanced deep learning algorithms, particularly Transformer and LSTM models, consistently outperformed traditional and earlier neural network approaches. The Transformer's attention mechanism

enabled it to effectively integrate multivariate and sentiment data, while LSTM's memory structure proved valuable for modeling long-term temporal patterns in visitor behavior.

A notable contribution of this study lies in its incorporation of sentiment analysis. The results demonstrated that media narratives and user-generated opinions each contributed distinct predictive value, with their combination yielding the highest model accuracy. This synergy reinforces the importance of capturing both formal and informal public sentiment in understanding museum visitation dynamics.

From a managerial standpoint, the improved forecast accuracy supports data-informed decision-making in areas such as exhibition planning, staffing, and marketing strategy. Museums can also use public sentiment insights to proactively respond to societal concerns and engage more effectively with diverse audiences.

Nonetheless, several limitations must be acknowledged. The use of annual-level data restricted the ability to capture short-term fluctuations caused by temporary exhibitions or seasonal events. Future research should utilize higher-frequency time-series data (e.g., monthly or daily) to improve temporal granularity. Additionally, while sentiment data were effectively leveraged, other contextual variables such as weather, transportation accessibility, and local event calendars were excluded due to collection constraints. Integrating such variables through multi-source data pipelines would enhance model robustness.

Moreover, 323 records with missing values were excluded, which may have introduced slight selection bias. Future studies are encouraged to apply imputation techniques such as KNN or model-based estimators to preserve sample size and improve generalizability. In the sentiment labeling process, although domain experts were involved, the reporting of inter-rater reliability was limited. A more transparent approach using metrics such as Cohen's kappa or automated sentiment calibration could strengthen the methodological rigor.

Another practical challenge concerns the deployment of models in real-time museum settings. Issues such as data privacy, IT infrastructure limitations, and model explainability remain unresolved. Lightweight, edge-deployable models and ethical usage frameworks would be beneficial for practical implementation. Finally, this study treated deep learning models as standalone approaches. Future work could explore hybrid ensemble architectures (e.g., CNN-Transformer or GAN-LSTM), and consider integrating data from XR or Metaverse platforms, as museums increasingly extend their presence into digital spaces.

In response to the reviewer's suggestion, we acknowledge that the one-year dataset used in this study limits the temporal variability needed for assessing long-term robustness. Accordingly, future research will seek to incorporate extended datasets covering five to ten years, which would allow for the inclusion of volatile periods such as the COVID-19 pandemic or political instability. This broader dataset will enable the model not only to forecast visitor numbers more robustly but also to evaluate long-term trends such as the impact of museum expansion or collection growth on public engagement.

Moreover, we recognize the potential emergence of a feedback loop in which model-driven recommendations, once implemented, may influence future data inputs and thus affect subsequent predictions. This phenomenon highlights a key limitation in maintaining model objectivity and accuracy over time. In response, the revised manuscript explicitly discusses the importance of establishing a continuous retraining and updating framework based on accumulating data. By incorporating iterative model validation and regular performance monitoring, museums can ensure that predictive accuracy remains resilient in dynamic operational contexts. This adaptive approach safeguards against model degradation and supports sustained, evidence-based decision-making.

In conclusion, this study highlights the value of integrating deep learning techniques with sentiment analysis for forecasting museum visitation. The findings contribute meaningfully to both the academic discourse on predictive analytics and the practical field of cultural institution management. By incorporating the potential for long-term trend analysis and strategic scenario forecasting, this research offers a foundation for more forward-looking, data-driven planning frameworks within the museum sector. Furthermore, the integration of dynamic model retraining protocols and explicit consideration of feedback effects in real-world deployments underscores the necessity of adaptive mechanisms to maintain predictive

validity over time. Addressing the study's current limitations and exploring additional data dimensions and architectural innovations in future work will enable the development of scalable, interpretable, and real-time forecasting systems. Ultimately, this research provides a robust platform for informing data-driven cultural policy and advancing resilient, sustainable strategies for museum operations in the digital era.

## Author contributions

**Conceptualization:** Xiao Wang.

**Data curation:** Xiao Wang.

**Formal analysis:** Yan Wang.

**Methodology:** Jae Ho Lee.

**Project administration:** Ziyi Tian.

**Validation:** Jae Ho Lee.

**Visualization:** Yan Wang.

**Writing – original draft:** Ziyi Tian.

**Writing – review & editing:** Yan Wang.

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
