## [Decision Letter · Decision Letter 0]

7 Apr 2025

Dear Dr. Wang,

Thank you for submitting your manuscript to PLOS ONE. After careful consideration, we feel that it has merit but does not fully meet PLOS ONE’s publication criteria as it currently stands. Therefore, we invite you to submit a revised version of the manuscript that addresses the points raised during the review process.

We look forward to receiving your revised manuscript.

Kind regards,

Muhammad Shahid Anwar

Academic Editor

PLOS ONE

Journal Requirements:

2. In your Methods section, please include additional information about your dataset and ensure that you have included a statement specifying whether the collection and analysis method complied with the terms and conditions for the source of the data.

4. For studies involving third-party data, we encourage authors to share any data specific to their analyses that they can legally distribute. PLOS recognizes, however, that authors may be using third-party data they do not have the rights to share. When third-party data cannot be publicly shared, authors must provide all information necessary for interested researchers to apply to gain access to the data. (https://journals.plos.org/plosone/s/data-availability#loc-acceptable-data-access-restrictions)

Reviewers' comments:

Reviewer's Responses to Questions

**Comments to the Author**

1. Is the manuscript technically sound, and do the data support the conclusions?

Reviewer #1: Yes

Reviewer #2: Yes

Reviewer #3: Yes

Reviewer #4: Yes

2. Has the statistical analysis been performed appropriately and rigorously?

Reviewer #1: Yes

Reviewer #2: Yes

Reviewer #3: Yes

Reviewer #4: Yes

3. Have the authors made all data underlying the findings in their manuscript fully available?

Reviewer #1: Yes

Reviewer #2: Yes

Reviewer #3: Yes

Reviewer #4: No

4. Is the manuscript presented in an intelligible fashion and written in standard English?

Reviewer #1: Yes

Reviewer #2: Yes

Reviewer #3: Yes

Reviewer #4: Yes

Reviewer #1: 1. How was the model validated to ensure it generalizes well across different museum environments or geographic locations?Could the authors provide further details on the external validation process, if any, such as using data from museums not included in the training dataset?

2. It would be beneficial to see a comparative analysis with traditional forecasting models. How do the proposed deep learning models perform in comparison to these more traditional approaches in terms of accuracy and computational efficiency?

3. The manuscript mentions the integration of sentiment analysis from news articles and comments. How significant is the impact of these external sentiment factors compared to internal data such as past visitor numbers or museum events?

Are there other external factors, such as local events or holidays, that could also be considered in the models?

4. Literature of the manuscript needs updation as suggested

a. Metaverse and XR for cultural heritage education: applications, standards, architecture, and technological insights for enhanced immersive experience

5. What are the main challenges in implementing this model in a real-world museum setting? Are there specific technological or data privacy barriers that need to be addressed? How does the model handle fluctuations in data quality, especially concerning real-time data integration?

Reviewer #2: - Data Preprocessing: Clarify how missing data were handled (e.g., imputation methods) and justify the exclusion of samples with missing information (n=323). A brief discussion on potential biases introduced by this exclusion would strengthen reproducibility.

- Sentiment Analysis Validation: While domain experts validated sentiment scores, include inter-rater reliability metrics (e.g., Cohen’s kappa) to quantify consensus among reviewers.

- Temporal Granularity: The study uses annual data (2023). Consider discussing the implications of this granularity and whether daily/monthly data could improve model performance.

- Statistical Analysis: The use of K-fold cross-validation (K=5) and multiple evaluation metrics (R², MSE, RMSE, MSLE, MAPE) is appropriate.

- Data Availability: The Data Availability Statement references the Korea Culture and Information Service (KCISA) and Naver-scraped text data.

Reviewer #3: 1.Research Problem and Novelty

The study proposes a deep learning-based model for predicting museum visitor numbers, integrating sentiment analysis data to improve prediction accuracy. However, the paper should further clarify its novelty compared to existing research. Is the primary contribution the integration of structured and unstructured data, or does the study also introduce innovations in model optimization and feature engineering? Strengthening this discussion in the introduction or literature review would enhance the paper's impact.

2.Methodological Rigor and Completeness

The paper employs multiple deep learning models (Transformer, LSTM, RNN, GAN, CNN) for comparative analysis and incorporates sentiment data for improved prediction. To enhance methodological clarity, consider elaborating on the following aspects:

What specific variations of Transformer and LSTM were used? Were hyperparameters optimized?

How were training and testing datasets split? Was cross-validation performed?

How was sentiment analysis conducted? What preprocessing techniques and sentiment scoring methods were applied? Providing such details would improve transparency and reproducibility.

3.Data Sources and Interpretability

The integration of structured museum data and unstructured opinion data is valuable, but additional clarifications are needed:

Is the dataset representative in terms of coverage and time span?

What criteria were used for selecting news articles and user comments? How was potential bias (e.g., media stance, extreme sentiment) addressed?

After incorporating sentiment data, how were model contributions quantified? Could techniques such as feature importance analysis or SHAP values be used for better interpretability?

4.Experimental Rigor and Validity

While the paper claims that Transformer and LSTM outperform traditional methods, a more detailed comparison would strengthen this argument:

Provide specific performance metrics (e.g., RMSE, MAE, R²) to demonstrate the improvement.

Did the study conduct significance tests to confirm that sentiment data integration yields statistically meaningful gains?

Were the impacts of news-based and user-comment-based sentiment data analyzed separately?

5.Enhancing the Conclusion Section

The conclusion is well-structured but could be refined further:

The discussion currently leans toward practical applications. Consider adding a section on the study’s limitations, such as data timeliness, sentiment analysis accuracy, and deep learning model interpretability.

Suggest future research directions, such as improving model explainability or integrating museum-specific operational data for personalized visitor forecasting.

6.Writing Quality and Presentation

The writing is generally clear, but some statements could be more precise. For example, instead of stating “improved prediction accuracy,” specify “achieved X% improvement compared to traditional methods.”

Ensure consistent terminology, such as standardizing “visitor prediction” and “visitor forecasting.”

Consider adding relevant visual aids (e.g., data processing flowcharts, model architecture diagrams, error comparison plots) to enhance readability.

7.Other details

(1) Based on the abstract, the study aims to develop a so-called museum visitor prediction model. However, it does not clearly specify the type of prediction—whether it forecasts visitor types, the timing of museum visits, or the types of museums being visited. Additionally, while the abstract discusses comparisons between different network algorithms, it does not explicitly state which algorithm was ultimately used in the developed model.

(2) In the introduction, since the study mentions the limitations of historical data, it should clarify what specific limitations exist. Additionally, it should explain in what ways the newly applied algorithms offer advantages over historical data-based methods.

(3) In Section 2.3, Research Questions, a well-structured academic paper typically addresses a single core research problem to ensure depth and coherence. The current three questions could be consolidated into one overarching issue: overcoming the limitations of historical data in accurately predicting museum visitor behavior through the introduction of new algorithms. The three listed questions seem more like titles for different stages of the research process rather than a singular core scientific problem.

8.Overall Assessment

This study presents a valuable approach to museum visitor prediction by integrating deep learning and sentiment analysis. However, improvements in data processing details, experimental design, result interpretation, and writing precision are needed to enhance the paper’s scientific rigor and clarity. Addressing these areas will significantly strengthen the manuscript and improve its chances of acceptance.

Reviewer #4: This paper presents a valuable contribution to the field of museum studies and sustainable management by developing a deep learning-based model for predicting museum visitor numbers. The study's innovative approach of integrating both structured museum-related data and unstructured external opinion data, through sentiment analysis of news articles and user comments, significantly enhances the predictive accuracy of visitor models. This comprehensive methodology offers insights into museum operations and demonstrates how predictive analytics can support sustainability in museums.

Of particular interest of this research is its operational relevance and awareness. By improving the accuracy of visitor predictions, the study demonstrates that museums could enhance plan for staffing, exhibition scheduling, public events, and facility management. In turn, this could support more effective resource allocation and potentially reduce operational costs which are currently of vital concern to museums globally.

Not only is economic sustainability a concern but many museums are also trying to remain socially sustainable in a dynamic changing society. The model’s incorporation of external factors such as media coverage and public opinion highlights the critical role that broader societal trends play in shaping visitor behaviour. This output is invaluable for museums in developing targeted marketing strategies and public engagement initiatives. The ability to understand the drivers behind visitor attendance empowers institutions to create more effective promotional campaigns, curate exhibitions that resonate with diverse audiences, and design educational programs that foster lasting connections with the public.

Ultimately, this research demonstrates that the social viability of museums is closely tied to their sustainability. The findings underline how museums can leverage data-driven insights to enhance their relevance and impact in an increasingly complex cultural landscape. This paper offers a forward-thinking approach to museum management, with clear implications for improving both operational efficiency and visitor engagement, making it an interesting read for museum professionals.

Some specific attributes of the paper worth noting include:

• Site Land Value an excellent inclusion in the assessed variables. Assessing whether a museum is adding value to a city and its people is greatly beneficial to understanding how the public engage with the space and how the museum fully (or not fully) utilises its prime location.

• It would be valuable to assess how the model's performance evolves when trained on a more extended dataset i.e. over a two-year, or more period. Ideally, a five-to-ten-year data set would be most beneficial and could include volatile periods such as the impact of the COVID-19 pandemic or disruptive political situations within a region. A longer time span would likely introduce greater variability in key variables like SLV and TAV, offering a more robust evaluation of the model’s predictive capabilities.

• With access to data across longer time spans, it could also be worthwhile to broaden the scope of the predicted variables to encompass future trends across the full dataset. This would help determine whether the model can effectively forecast potential benefits of expanding the museum or its collections.

• A key limitation lies in evaluating how the implementation of model-generated recommendations may influence user behaviour. Once these recommendations are applied, the resulting changes become part of the data-generating process, complicating efforts to separate cause and effect within the same predictive framework. For instance, when the model produces a prediction and the museum adjusts its parameters accordingly, those adjustments are reflected in the data fed back into the system. As a result, the model’s future inputs will comprise both historical data and the institution’s response to prior outputs, potentially creating a feedback loop over time. This suggests that the predictive value of such models may be finite unless mechanisms are in place to regularly update and retrain the model as new data becomes available. Continual retraining would help ensure that the model remains responsive to evolving user behaviours and institutional changes, thereby maintaining the integrity and relevance of its predictions.

• It was interesting to see that sentiment analysis in this paper used controversy as a negative. As far as museum numbers go, controversy isn’t a negative thing. Active discourse and critique, in my experience, is encouraged in museum spaces. Museums are often viewed as places that provide all the information related to a topic without bias, allowing for discussions and open conversation. However, the authors do allow for change in sentiment analysis locally so this would be adapted to suit each country’s (or museum’s) outputs.

**Do you want your identity to be public for this peer review?** For information about this choice, including consent withdrawal, please see our Privacy Policy

Reviewer #1: **Yes: ** Niyaz Ahmad Wani

Reviewer #2: No

Reviewer #3: No

Reviewer #4: No

---

## [Author Response · Author response to Decision Letter 1]

6 Aug 2025

Dear Editors,

Thank you very much for your time and thoughtful evaluation of our manuscript. We sincerely appreciate the valuable comments and suggestions provided by the reviewers and editorial team, which have helped us to significantly improve the quality and clarity of our paper.

Due to the length and detailed nature of our responses, we have prepared a separate document addressing each comment point-by-point. This response file is included as a supplementary attachment in our submission.

We hope that our revisions and detailed replies sufficiently address all concerns and meet the expectations of the reviewers and the editorial board. Should any additional clarification be needed, we would be happy to provide it.

Thank you again for your consideration.

---

## [Decision Letter · Decision Letter 1]

15 Oct 2025

Enhancing Museum Visitor Forecasting Using Deep Learning and Sentiment Analysis: A Transformer-Based Approach for Sustainable Management

PONE-D-25-07092R1

Dear Dr. Wang,

We’re pleased to inform you that your manuscript has been judged scientifically suitable for publication and will be formally accepted for publication once it meets all outstanding technical requirements.

Kind regards,

Muhammad Shahid Anwar

Academic Editor

PLOS ONE

Additional Editor Comments (optional):

Reviewers' comments:

Reviewer's Responses to Questions

**Comments to the Author**

Reviewer #1: All comments have been addressed

2. Is the manuscript technically sound, and do the data support the conclusions?

Reviewer #1: Yes

3. Has the statistical analysis been performed appropriately and rigorously?

Reviewer #1: Yes

4. Have the authors made all data underlying the findings in their manuscript fully available?

Reviewer #1: Yes

5. Is the manuscript presented in an intelligible fashion and written in standard English?

Reviewer #1: Yes

Reviewer #1: All the comments have been done by the authors in a better way now.

My Recommendation is Accept

Best of Luck!

**Do you want your identity to be public for this peer review?** For information about this choice, including consent withdrawal, please see our Privacy Policy

Reviewer #1: No

---

## [Editor Report · Acceptance letter]

PONE-D-25-07092R1

PLOS ONE

Dear Dr. Wang,

I'm pleased to inform you that your manuscript has been deemed suitable for publication in PLOS ONE. Congratulations! Your manuscript is now being handed over to our production team.

Kind regards,

on behalf of

Professor Muhammad Shahid Anwar

Academic Editor

PLOS ONE